# Fusion Networks of CNN and Transformer with Channel Attention for Accurate Tumor Imaging in Magnetic Particle Imaging

**Yaxin Shang** [1], **Jie Liu** [1,*] **and Yueqi Wang** [2,3,*]

1   School of Computer and Information Technology, Beijing Jiaotong University, Beijing 100044, China;
    20112039@bjtu.edu.cn
2   CAS Key Laboratory of Molecular Imaging, Beijing Key Laboratory of Molecular Imaging,
    Institute of Automation, Beijing 100190, China
3   University of Chinese Academy of Sciences, Beijing 100080, China
*   Correspondence: jieliu@bjtu.edu.cn (J.L.); yueqi.wang@ia.ac.cn (Y.W.)

**Simple Summary:**   Accurate tumor localization is essential for effective clinical diagnosis and treatment. However, traditional magnetic particle imaging (MPI) algorithms struggle to precisely locate tumors, resulting in challenges when quantifying them. This study aims to address the issue of precise tumor localization in MPI through the application of a deep learning approach. By integrating Convolutional Neural Network (CNN) and Transformer modules, the goal is to improve image quality and enhance the accuracy of tumor quantification in MPI images. The research utilizes a combination of CNN and Transformer modules to capture both global and local features within MPI images. Through the application of deep learning techniques, the study seeks to remove blurry artifacts from reconstructed images, ultimately may help improve the precision of tumor localization and quantification in MPI. This approach holds significant potential for advancing MPI technology and introducing novel methodologies for future medical imaging research. Additionally, the validation of transfer learning on authentic MPI images may enhance the overall accuracy and reliability of MPI image reconstruction.

**Abstract:** Background: Magnetic Particle Imaging (MPI) is an emerging molecular imaging technique. However, since X-space reconstruction ignores system properties, it can lead to blurring of the reconstructed image, posing challenges for accurate quantification. To address this issue, we propose the use of deep learning to remove the blurry artifacts; (2) Methods: Our network architecture consists of a combination of Convolutional Neural Network (CNN) and Transformer. The CNN utilizes convolutional layers to automatically extract pixel-level local features and reduces the size of feature maps through pooling layers, effectively capturing local information within the images. The Transformer module is responsible for extracting contextual features from the images and efficiently capturing long-range dependencies, enabling a more effective modeling of global features in the images. By combining the features extracted by both CNN and Transformer, we capture both global and local features simultaneously, thereby improving the quality of reconstructed images; (3) Results: Experimental results demonstrate that the network effectively removes blurry artifacts from the images, and it exhibits high accuracy in precise tumor quantification. The proposed method shows superior performance over the state-of-the-art methods; (4) Conclusions: This bears significant implications for the image quality improvement and clinical application of MPI technology.

**Keywords:** magnetic particle imaging; convolutional neural network; transformer; tumor imaging; accurate quantification





# 1. Introduction

Magnetic Particle Imaging (MPI) is an emerging non-radiative and non-invasive biomedical imaging technique that employs magnetic nanoparticles (MNPs) injected into the biological body to generate high-resolution images [1]. MPI offers exceptional spatial resolution [2,3], enabling imaging at the millimeter level, thereby facilitating more accurate lesion localization and cellular-level biological process studies. As MPI does not employ X-rays or other forms of radiation, it ensures safety for both patients and operators. Due to these advantages, MPI holds potential for detecting anomalies within the biological body, studying organ functions, and conducting molecular-level biomedical research [4–9].

An MPI imaging system primarily consists of three components: the selection field, the drive field, and the receiving coils. The selection field is a static magnetic field formed by two like-polarity magnets, generating a Field Free Region (FFR) at the center [10]. The drive field is an applied alternating magnetic field that moves the FFR along a predetermined trajectory. The basic principle of MPI imaging is illustrated in Figure 1 [11], where the black curves in Figure 1a,b represent the magnetization curve of MNPs, the green curve represents the applied alternating magnetic field, the red curve indicates the resulting response magnetic field signal, and the lower right depicts the frequency spectrum of the response signal. When MNPs are within or near the FFR, they generate detectable signals under the influence of the drive field. In regions outside the FFR, the MNPs are in a nearly saturated state, exhibiting minimal changes in magnetization over time and resulting in negligible induced signals. Figure 2 represents MPI imaging process. Firstly, tumor mice labeled with MNPs is introduced into the MPI system. Secondly, after scanning the mice on the MPI platform, MPI signals are generated at the location of the tumor. Finally, through image reconstruction, the system can produce imaging results and accurately locate the tumor position.

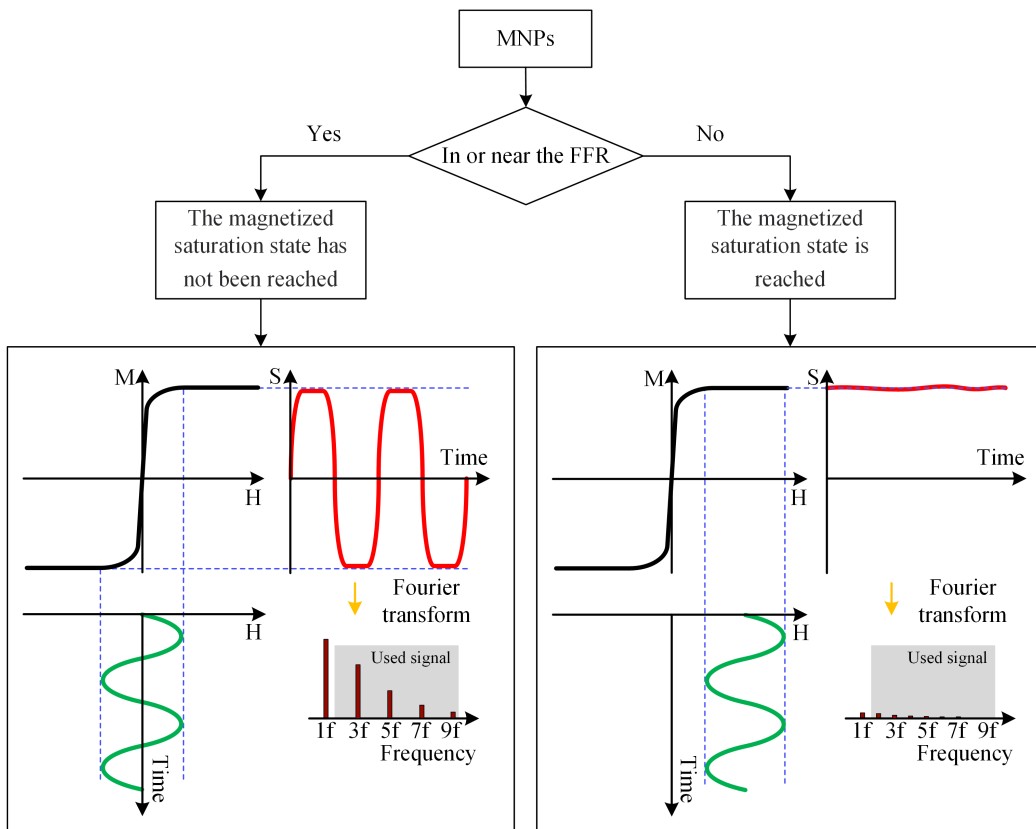

**Figure 1.** Basic principles of MPI imaging. (**a**) The response signal of the MNPs in the unsaturated state. (**b**) Saturation state of MNPs signal.

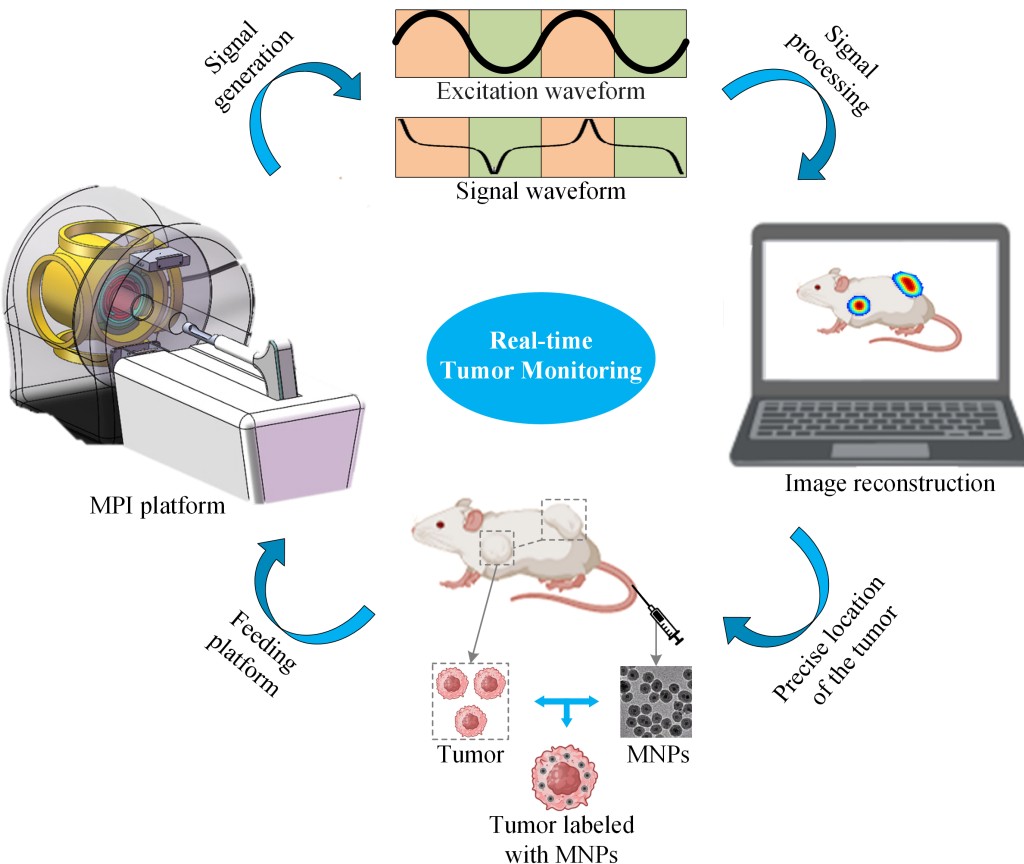

**Figure 2.** Schematic diagram of MPI imaging system.

In MPI, two primary reconstruction methods exist: X-space [12,13] and system matrix (SM) reconstruction [14]. Both methods are employed to reconstruct images of the spatial distribution of magnetic particles within a target area from collected magnetic signal data. The key to system matrix-based reconstruction in MPI lies in constructing the system matrix. The SM describes the mapping between spatial concentration distribution and induced voltage signals, composed of the Fourier components of time signals produced by each magnetic nanoparticle at all possible spatial positions. The SM is constructed through measurements of MNPs' spatial movement along specific trajectories or through mathematical modeling. However, the construction of the SM is time-consuming and memory-intensive [15]. On the other hand, the X-space method separates image reconstruction from convolution, allowing for modular imaging and faster reconstruction [16]. Nevertheless, X-space reconstruction introduces artifacts that can impact the localization accuracy of MNPs. Previous research has often employed traditional deconvolution methods to mitigate these artifacts, but deconvolution can amplify noise and introduce additional image artifacts [17,18].

As a prominent branch of artificial intelligence, deep learning continues to make significant breakthroughs in various fields [19,20]. Within this landscape, the field of artifact removal holds a crucial place as an important application of deep learning. In recent years, researchers have shown increasing interest and dedication to this area. Artifact removal, situated within image processing and computer vision domains, involves in-depth exploration and mitigation of artifacts, noise, and other adverse factors present in images [21,22]. The overarching goal of this domain is to leverage deep learning techniques to accurately identify and effectively eliminate artifacts from images, thereby enhancing image quality and usability. In the past, artifacts posed significant challenges in medical imaging, leading to compromised reliability and accuracy. Nevertheless, the integration of deep learning into the field of medical image artifact removal has yielded substantial progress [23–25].

Lin et al. [26] utilized an end-to-end trainable dual-domain network (DuDoNet) to eliminate metal artifacts in CT images, enhancing image contrast and restoring the true structure of scanned subjects. Similarly, Luthra et al. [27] proposed the use of transformers for medical image denoising while preserving edge information. In recent years, deep learning have received interest in MPI as a powerful alternative [28,29]. Shang et al. [30] proposed a cost-effective method called the fusing dual-sampling network (FDS-MPI), which effectively enhances the resolution of low-gradient MPI. Gungor et al. [31] tackled the issue of time-consuming system matrix calibration by utilizing the transformers for system matrix super-resolution (TranSMS). Wu et al. [32] addressed the sparse-view problem in projection MPI by introducing a projection generative network (PGNet), which generates novel projections. Gungor et al. [33] designed a new physics-driven method for MPI reconstruction based on a deep equilibrium model with learned data consistency (DEQ-MPI). However, there is a limited amount of research dedicated to enhancing MPI image quality by specifically addressing the removal of blurred artifacts and the recovery of edge information.

In this study, leveraging the physical characteristics of MPI imaging, we employ a cascaded CNN-Transformer deep learning network model to address the issue of image blurriness in the X-space method. he structural composition is fundamentally characterized by three key elements: a dual-branch convolution module, an efficient transformer module, and a feature fusion with channel attention module. The dual-branch convolution module operates two distinct convolutional neural networks in parallel. These branches are designed to extract diverse and complementary features from the input data. Additionally, the transformer module utilizes efficient multi-head self-attention (EMHA) and multi-layer perceptions (MLPs) to capture intricate patterns and meaningful features within the input sequence. Furthermore, the feature fusion module leverages convolutional layers enriched with attention mechanisms, particularly channel attention, to integrate and refine the features obtained from the preceding modules. The results demonstrate the efficacy of the CNN-Transformer network in effectively addressing image blurriness in MPI, leading to improved spatial localization capabilities and achieving clear imaging. This advancement holds promise for enhancing the potential of MPI in spatial precision and accuracy. The main innovations and contributions of our research can be summarized as follows:

(1)  Structural Composition: We propose a novel deep learning network model that combines a dual-branch convolution module, an efficient transformer module, and a feature fusion with channel attention module to address image blurriness in MPI.
(2)  Improved Image Quality: The proposed method effectively mitigates blurring artifacts, resulting in improved spatial localization capabilities and clearer imaging.
(3)  Potential Impact on MPI: Our approach has the potential to enhance the capabilities of MPI as a non-invasive imaging technique by improving spatial precision and accuracy. This paper is structured as follows: Section 2 presents the materials and methods used for our experiments, followed by Section 3, which details the results obtained. In Section 4, we discuss the implications of our findings, and finally, in Section 5, we conclude with a summary of our key contributions and potential future directions for this research.

## 2. Materials and Methods

### 2.1. Dataset

#### 2.1.1. Simulation Data

Ensuring an ample and diverse dataset to train our model was a foundational aspect of our research. The structure of simulation data was hand-drawn, with varying sizes, shapes, and numbers of tumor features. Each feature was individually designed and created by our team using custom drawing tools. This ensured that the tumors exhibited diverse characteristics and variations in size, shape, and number. This dataset consisted of 400 original binary images, diversified by allocating distinct grayscale domains to simulate varying concentrations of MNPs. To introduce particle concentration gradients, we utilized

different blur kernels to generate gradient grayscale values. Additionally, we increased the diversity of our dataset by incorporating flips and rotations to the binary images, grayscale images, and gradient grayscale images. The image size is 50 × 50 pixels. This holistic approach culminated in an extensive dataset, comprising a total of 8000 images. Among these, 7000 images were earmarked for training purposes, while the remaining 1000 images served as the test set. We have made our data publicly available on https://github.com/kutuy123/MPI-Phantom-dataset. Our research was the utilization of the Langevin equation to meticulously simulate the magnetic responses of MNPs in the MPI context. The received MPI signal $s(t)$ can be expressed as in [17]:

$$s(t) = \frac{d}{dt}\sigma_s(\mathbf{r})M(\mathbf{r}, t)d\mathbf{r}, \tag{1}$$

where:

$$M(\mathbf{r}, t) = m_0 \rho(\mathbf{r}) L\Big[\frac{\|\mathbf{G}(\mathbf{r}_s(t) - \mathbf{r})\|}{H_{sat}}\Big] \times \frac{\mathbf{G}(\mathbf{r}_s(t) - \mathbf{r})}{\|\mathbf{G}(\mathbf{r}_s(t) - \mathbf{r})\|}, \tag{2}$$

where $\sigma_s$ is the sensitivity of the receiver coil; $M(\mathbf{r}, t)$ is the magnetization density at time $t$ and position $\mathbf{r}$; $m_0$ is the magnetic moment $\rho$ is the nanoparticle concentration; $L$ is the Langevin function; $\mathbf{G}$ is the gradient intensity. The actual reconstruction of MPI images was conducted through the application of the X-space method, grounded in the principles of X-space theory. Employing the capabilities of MATLAB (Mathworks, Natick, MA, USA), we generated simulated MPI images tailored to the x-z plane. To ensure a seamless integration of authentic noise disturbances inherent to MPI imaging, we turned to a commercially available MPI device—the MOMENTUM CT by Magnetic Insight Inc., based in Alameda, CA, USA. This device facilitated the collection of genuine noise images, capturing the intricate nuances of noise patterns arising during actual MPI imaging operations. By employing the device, we gathered genuine noise images, guaranteeing that the captured noise patterns accurately reflect the real noise experienced during the MPI imaging procedure [34]. Regarding the spatial characteristics of the noise, it was spatially varying in nature. This means that the noise patterns differed across different regions or areas within the images. Following this, we cropped 10 noisy images and added them to the native MPI images, thereby creating a dataset with signal-to-noise ratios (SNR) ranging from 5 to 15 dB. Table 1 lists MPI simulation parameters.

**Table 1.** The MPI simulation parameters.

| Symbol | Parameters | Value | Unit |
|--------|-----------|-------|------|
| $\mu_0$ | Permeability of vacuum | $4\pi \times 10^{-7}$ | $NA^{-2}$ |
| $D$ | Nanoparticle diameter | 20 | nm |
| $T_0$ | Kelvin temperature | 293 | K |
| $k_B$ | Boltzmann constant | $1.28 \times 10^{-23}$ | $JK^{-1}$ |
| $m_0$ | Magnetic moment | $6.75 \times 10^{-18}$ | $Am^2$ |
| $G$ | Gradient intensity | 6 | $T/m^{-1}\mu_0^{-1}$ |
| $\sigma$ | SNR | 5–15 | dB |

## 2.1.2. Experimental Data

To validate the feasibility of our method, we conducted experiments using 100 experiments data. The phantoms were created using 3D printing technology and were made from resin materials. Each phantom had dimensions of 3 cm × 3 cm. The height is 0.8 cm. The depth of the features within the phantoms is 0.6 cm. The shape of the features is randomly generated. To introduce the MNPs into the phantoms, we used commercially avaliable MNPs, Perimag® 102-00-132 (micromod Partikeltechnology GmbH, Rostock, Germany). The MNPs were inserted into the phantoms at a concentration of 1 mg/mL. The phantoms were then scanned using a commercial MPI scanner (MOMENTUM CT, Magnetic Insight

Inc., Alameda, CA, USA). The field of view (FOV) size is 4 × 4 cm. The scan mode is the high resolution mode (G = 5.7 T/m$^{-1}\mu_0^{-1}$). The scanner utilized its proprietary software to generate the native image files from the acquired MPI data.

## 2.2. Network Architecture

The workflow of this study, illustrated in Figure 3, unfolds in three main phases: data preprocessing, network training, and MPI image deblurring. In the data preprocessing phase, the dataset undergoes augmentation operations, including inversion and rotation, to enrich its diversity. Subsequently, this augmented dataset is subjected to normalization for consistency. Once the training data is prepared, it is fed into our model to initiate the training process, adjusting model parameters for optimization. Upon completion of training, the trained model can be employed to input MPI images afflicted with artifacts, thereby generating predictions of artifact-free, clear images.

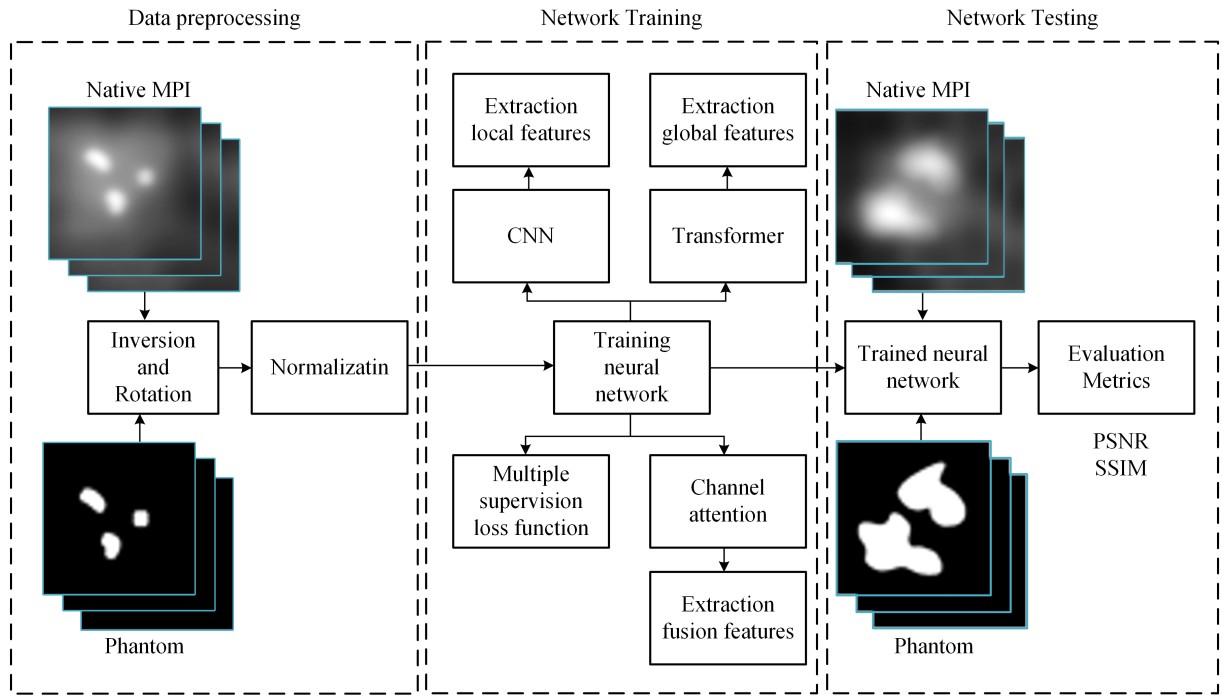

**Figure 3.** Illustrative workflow of the study.

The network architecture is shown in Figure 4. Our proposed architecture leverages the inherent physical attributes of MPI imaging, presenting a cascaded CNN-Transformer deep learning network model. The proposed network comprises three main components: a dual-branch convolution module, a transformer module, and a feature fusion module. The dual-branch convolution module is designed to capture local features in MPI images. It consists of two parallel CNNs. One network extracts relevant information, while the other focuses on removing unwanted artifacts, effectively addressing image blurriness issues in MPI images. The transformer module is designed to extract global features and mitigating overall image non-uniformity. It contains convolutional projection, EMHA, and MLP components. The convolutional projection module utilizes depth-wise separable convolution (DWSC) to extract query, key, and value components for each marker. These projections are then processed by the EMHA module, followed by the extraction of global features using the MLP. This module primarily focuses on alleviating global image non-uniformity. The feature fusion module predominantly consists of convolutional layers and effectively integrates local and global features using attention mechanisms.

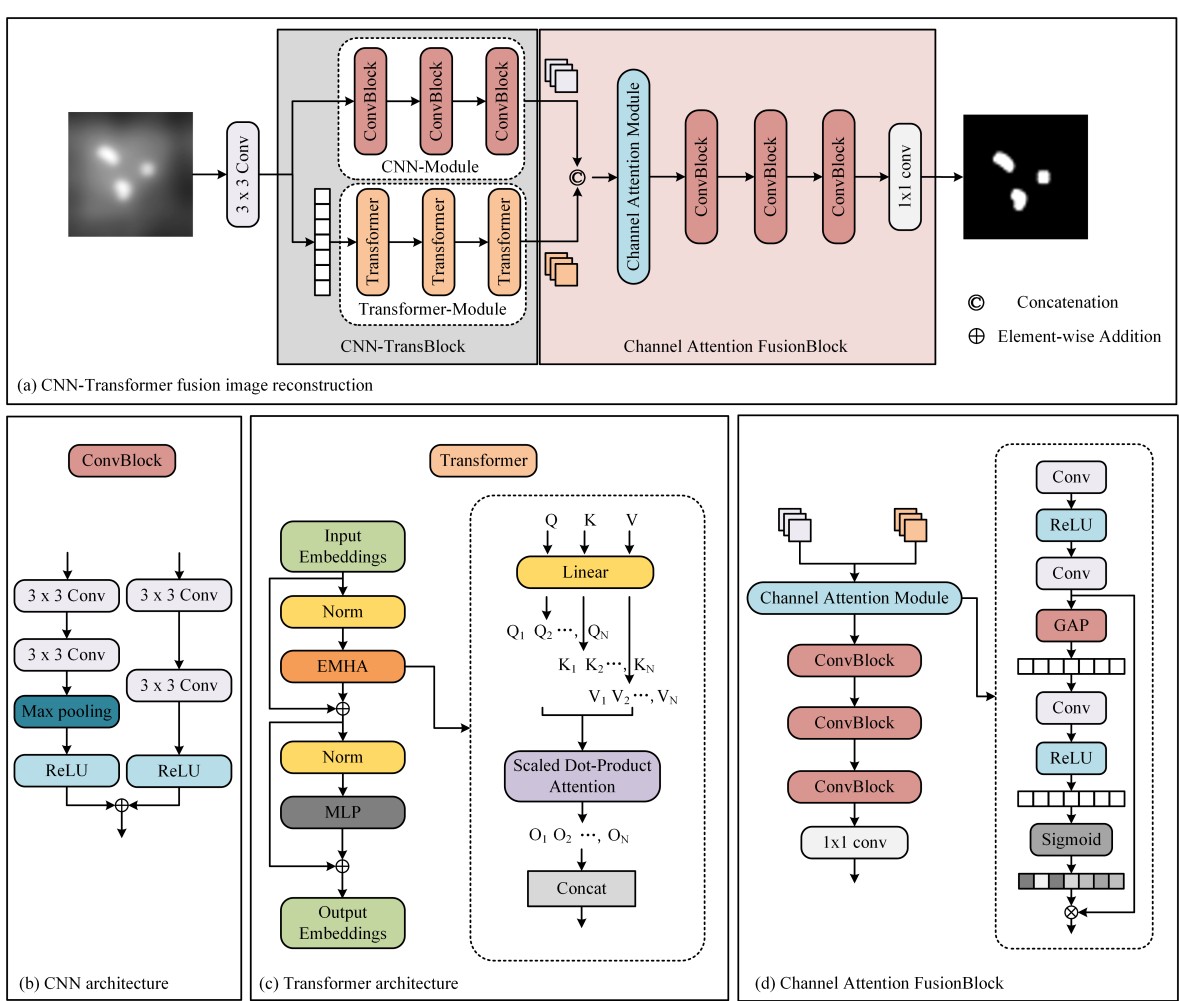

**Figure 4.** The architecture of our method.

### 2.2.1. Convolutional Neural Network (CNN) Module

The CNN module leverages twin convolutional networks to extract local features from MPI images, with each independent branch network serving a distinct purpose. Here, we adopt a dual-branch network architecture, where one branch consists of convolutional layers, ReLU layers, and downsampling layers. The other branch includes convolutional layers and ReLU layers. The purpose of this approach is to preserve the essential features of MPI while simultaneously filtering out unwanted artifacts. Subsequently, we combine the two parallel convolutional networks by adding them together, ensuring that the images are resized to the same dimensions before addition. In the CNN module, a total of three convolutional networks are utilized. We denote the feature that the CNN module extracts as $\mathbf{U_c}$.

### 2.2.2. Transformer Module

To extract the comprehensive spatial characteristics of MPI images, we employ the Transformer module. This module is composed of three essential components: convolutional projection, EMHA, and MLP. The convolutional projection module plays a pivotal role by employing DWSC to extract crucial elements such as query ($Q$), key ($K$), and value ($V$) values for each marker. This can be written as the following equation:

$$Q, K, V = Flatten(DWSC((\mathbf{I})), \tag{3}$$

where $\mathbf{I}$ is the input of the transformer module. This extraction process underpins the foundation of EMHA, facilitating the establishment of meaningful relationships between different parts of the image. In greater detail, the convolutional projection commences by

implementing DWSC, which serves as an effective mechanism to parse out the fundamental *Q*, *K*, and *V* components. These components, integral for the subsequent EMHA module, enable the establishment of vital associations between distinct elements within the image. The EMHA element, constituting the core of the Transformer module, is instrumental in capturing the intricate interdependencies within the image's global context. The EMHA operation concurrently considers multiple "heads", each concentrating on distinct aspects of the image, effectively synthesizing a comprehensive representation of global spatial relationships. This, in turn, enriches the model's understanding of the overall image composition. The process is defined as follows:

$$\mathbf{Q}, \mathbf{K}, \mathbf{V} = linear_i(Q, K, V), i = 1, 2, \ldots, N, \tag{4}$$

where $linear_i()$ is the parametric representation associated with the *i*-th head for query, key, and value. each head is applied with a Scaled Dot-Product Attention (SDPA) operation:

$$\mathbf{O}_i = softmax(\frac{(\mathbf{Q}_i(\mathbf{K}_i)^T)}{\sqrt{H \times W \times C}})(\mathbf{V}_i)^T, \tag{5}$$

To generate the complete output feature, we concatenate all of the outputs ($\mathbf{O}_1$, $\mathbf{O}_2$, …, $\mathbf{O}_N$) produced by SDPA. This process combines the attention maps from each head of the Transformer network to create a comprehensive representation of the input sequence.

$$\mathbf{O} = concat(\mathbf{O}_i), i = 1, 2, \ldots, N, \tag{6}$$

where, *concat* represents the channel fusion of features. The features extracted by the EMHA module are denoted as $\mathbf{U}_{EMHA}$ and can be represented by the following equation:

$$\mathbf{U}_{EMHA} = \mathbf{O} + \mathbf{I}_2, \tag{7}$$

Furthermore, the MLP within the Transformer module contributes to refining the extracted global features. The MLP applies a sequence of densely connected layers to distill and consolidate these features, enabling a more refined understanding of the image's holistic context. The features extracted by the Transformer module are represented as $U_t$ and can be described by the following equation:

$$\mathbf{U}_t = MLP(Norm(\mathbf{U}_{EMHA})) + \mathbf{U}_{EMHA}, \tag{8}$$

2.2.3. Channel Attention Fusion Module

The feature fusion module plays a pivotal role in consolidating the features extracted by the convolutional and Transformer networks. This module harmonizes the local features from the convolutional network with the global features from the Transformer network, creating a multi-channel information representation.

$$\mathbf{U}_f = F(concat(\mathbf{U}_c, \mathbf{U}_t)), \tag{9}$$

where $\mathbf{U}_f$ represents the fusion teature, *F* is the feature fusion module. To further enhance the discriminative power of the fused features, we employ a channel attention mechanism. The Global Average Pooling (GAP) plays a crucial role in the channel attention mechanism. By introducing the GAP operation, the model can automatically learn the correlation and importance among different channels and adjust the representational capacity of the feature map based on these learned weights. This mechanism selectively emphasizes the informative channels while suppressing irrelevant or noisy channels within the fused feature representation.

$$\mathbf{F} = sigmoid(\mathbf{W}_2 * ReLU(\mathbf{W}_1 * \mathbf{U}_f)) \otimes \mathbf{U}_f, \tag{10}$$

where $\mathbf{W}_1$ and $\mathbf{W}_2$ is the convolution kernel parameter; $\otimes$ is Multi-plication of elements. $\mathbf{F}$ is final fusion teature.

### 2.2.4. Multi-Supervisory Loss Function

The utilization of a multi-supervisory loss function is a cornerstone of our approach, amalgamating the losses from each individual module. The loss function employed is the Mean Squared Error (MSE), and it is defined as follows:

$$MSE(X,Y) = \frac{1}{M \times N} \sum_{i=1}^{M} \sum_{j=1}^{N} \|X(i,j) - Y(i,j))\|_2^2, \tag{11}$$

where, $X$ and $Y$ respectively represent the predicted and ground truth artifact-free MPI images, with dimensions $M \times N$. The envisaged multi-supervisory loss function is crafted as follows:

$$Loss = \lambda_a \cdot MSE(X_a, Y_a) + \lambda_b \cdot MSE(X_b, Y_b) + \lambda_f \cdot MSE(X_f, Y_f), \tag{12}$$

where, $X_a$ and $Y_a$ represent the predicted and ground truth artifact-free MPI images from the dual-branch convolution module, while $X_b$ and $Y_b$ correspond to those from the Transformer module. Similarly, $X_f$ and $Y_f$ symbolize the predicted and ground truth artifact-free MPI images from the feature fusion module. The final loss is an amalgamation in the form of a weighted sum, where the weights are denoted as $\lambda_a$, $\lambda_b$, and $\lambda_f$. This multi-supervisory loss strategy circumvents the challenge of gradient vanishing during the backpropagation process, bolstering the robustness of the algorithm.

### 2.2.5. Transfer Learning

To employ pre-trained models for transfer learning, we opted for a deep neural network that had been trained on a simulation dataset as our base model. The weights of this base model were then loaded into our target task model as initialization parameters. Subsequently, we froze the weights of the base model and exclusively trained the last few layers or introduced additional layers in the target task model. This strategic approach allowed us to effectively exploit the feature representations acquired by the base model through training on the extensive dataset, consequently enhancing the performance and generalization capabilities of our model.

### 2.2.6. Evaluation Metrics

For a quantitative assessment, we used two metrics: peak signal to noise ratio (PSNR) [35] and structural similarity index measure (SSIM) [36]. PSNR is estimated as:

$$PSNR = 10 log_{10}(\frac{255^2}{MSE}), \tag{13}$$

where MSE is estimated as:

$$MSE = \frac{1}{m} \sum_{i=1}^{m} (X_i - \widehat{X_i})^2, \tag{14}$$

where $X_i$ and $\widehat{X_i}$ are the means of the real and estimated MPI images, respectively, and $m$ represents the number of images. SSIM is estimated as:

$$SSIM = \frac{(2\mu_X \mu_{\hat{X}} + t_1)(2\sigma_{X\hat{X}} + t_2)}{(\mu_X^2 + \mu_{\hat{X}}^2 + t_1)(\sigma_X^2 + \sigma_{\hat{X}}^2 + t_2)}, \tag{15}$$

where $\mu_X$ and $\mu_{\hat{X}}$ are the means of real and estimated MPI images, respectively. $t_1$ and $t_2$ are constants, $\sigma_X^2$ and $\sigma_{\hat{X}}^2$ represent the variances, and $\sigma_{X\hat{X}}$ is the covariance between $X$ and $\hat{X}$.

## 3. Results

### 3.1. Simulation Data Analysis

To validate the efficacy of our proposed approach, we conducted a series of experiments using diverse datasets and compared our method with conventional techniques such as blind deconvolution, Lucy deconvolution, and Wiener filtering. In addition, we also contrasted our method with the learning-based approach (FDS-MPI [30]). We conducted validation on the simulation dataset, selecting three different tumor shapes, each containing simulated instances of two tumors. As depicted in Figure 5, the first, third, and fifth rows illustrate the reconstruction outcomes for various simulation data, while the second, fourth, and sixth rows represent the corresponding error maps. The findings revealed that, irrespective of the tumor's shape, our method adeptly restored the precise boundary contours of the simulation data. This capability holds critical significance for accurate tumor localization. Additionally, we quantified the results, and the quantitative outcomes are presented in Table 2. The PSNR of traditional methods did not exceed 20, and the SSIM did not exceed 0.5. Our method achieved a PSNR of 27.9796 and an SSIM of 0.8062.

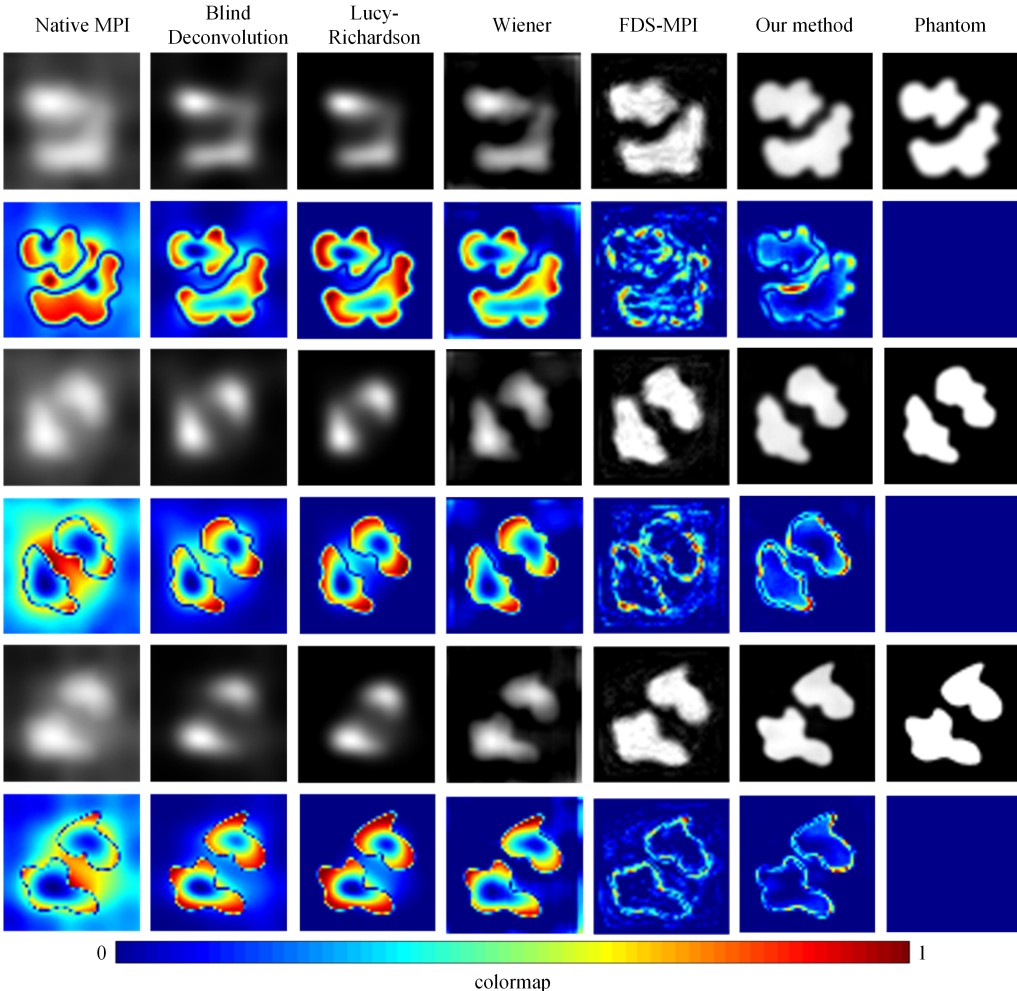

**Figure 5.** Visualization results of different methods with simulation dataset. The reconstruction outcomes for various simulation data are illustrated in the first, third, and fifth rows, while the corresponding error maps are represented in the second, fourth, and sixth rows.

**Table 2.** The metrics results of different methods.

| Metrics | Native MPI | Blind Deconvolution | Lucy-Richardson | Wiener | FDS-MPI | Our Method |
|---------|-----------|---------------------|-----------------|--------|---------|-----------|
| PSNR | 13.9062 ± 1.0367 | 11.1576 ± 1.5498 | 17.0005 ± 2.8273 | 18.2658 ± 3.2391 | 23.6107 ± 4.0975 | 27.9796 ± 2.4312 |
| SSIM | 0.1806 ± 0.0291 | 0.1668 ± 0.0938 | 0.3978 ± 0.1002 | 0.4689 ± 0.0850 | 0.6291 ± 0.09913 | 0.8062 ± 0.0241 |

During MPI imaging, the accurate quantification of low-concentration targets is often challenging when multiple targets with varying concentrations are present in the field of view. In our study, we simulated five concentration ratios: 1:0.2, 1:0.4, 1:0.6, 1:0.8, and 1:1. Each ratio was applied to two tumor targets, and MPI imaging was performed. The resulting images were analyzed using conventional methods such as Blind deconvolution, LucyRichardson, and Winear algorithms, as well as our proposed deep learning algorithm. In Figure 6, the first column of the figure displays the original MPI images, while the second to fourth columns present the results obtained from the aforementioned conventional methods. The fifth column showcases the outcomes of our proposed approach, while the sixth column presents the ground truth concentration distribution. Notably, the traditional methods struggled to accurately reconstruct the target with concentration differences, particularly the Blind and Lucy methods. Conversely, our deep learning algorithm demonstrated superior performance in restoring the original concentrations of both targets, thereby may help facilitate precise tumor localization. To further evaluate the performance of the different methods, we conducted a quantitative analysis. The result of Table 3 demonstrates the substantial improvements achieved by our proposed method across two metrics: PSNR and SSIM. Moreover, irrespective of the concentration ratio, our method consistently outperformed the other techniques, particularly in cases with no concentration difference between the two targets.

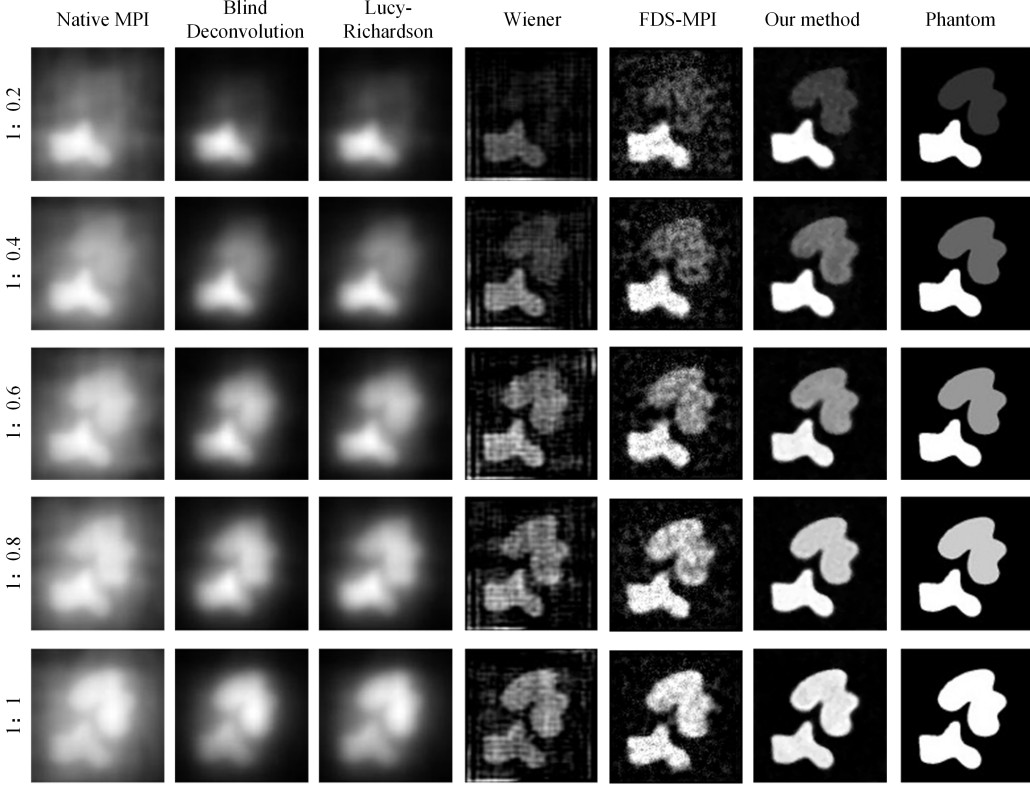

**Figure 6.** Visualization results of multiple targets with varying concentrations.

**Table 3.** The metrics results of multiple targets with varying concentrations.

| Concentration Ratio | Metrics | Native MPI | Blind Deconvolution | Lucy-Richardson | Wiener | FDS-MPI | Our Method |
|---|---|---|---|---|---|---|---|
| 1:0.2 | PSNR | 8.3267 | 11.4971 | 11.3208 | 11.1897 | 19.2972 | 22.0169 |
| | SSIM | 0.1248 | 0.1245 | 0.1197 | 0.2959 | 0.5108 | 0.6097 |
| 1:0.4 | PSNR | 10.2323 | 15.2176 | 15.4426 | 12.8129 | 20.5721 | 25.5540 |
| | SSIM | 0.1320 | 0.2012 | 0.2281 | 0.2923 | 0.3720 | 0.4247 |
| 1:0.6 | PSNR | 9.2481 | 14.0769 | 14.0875 | 13.9752 | 21.2109 | 25.6356 |
| | SSIM | 0.1283 | 0.1538 | 0.1568 | 0.2856 | 0.4019 | 0.4890 |
| 1:0.8 | PSNR | 8.8087 | 13.1996 | 13.0744 | 14.7216 | 21.0128 | 25.5039 |
| | SSIM | 0.1251 | 0.1408 | 0.1362 | 0.2833 | 0.4209 | 0.5171 |
| 1:1 | PSNR | 8.4024 | 12.3050 | 12.2112 | 13.1736 | 22.1980 | 25.3194 |
| | SSIM | 0.1262 | 0.1283 | 0.1204 | 0.2811 | 0.4381 | 0.5053 |

### 3.2. Ablation Study

We conduct extensive ablation studies to evaluate the individual contributions of the CNN module, Transformer module, and channel attention module. We conduct experiments on a testing dataset of MPI images to evaluate the performance of each module and their combination.

CNN Module: We first evaluate the performance of our proposed approach using only the CNN module. As shown in the Table 4, the PSNR and SSIM of the reconstructed image are 25.6309 and 0.6623.

**Table 4.** The results of ablation study. Each rows represents the inclusion of a different module (CNN, Transformer, Feature, Channel Attention), and a checkmarks indicates the inclusion of that module.

| CNN | Transformer | Feature Fusion | Channel Attention | PSNR | SSIM |
|---|---|---|---|---|---|
| ✓ | | | | $25.6309 \pm 2.8278$ | $0.6623 \pm 0.0321$ |
| | ✓ | | | $26.1242 \pm 3.9173$ | $0.7091 \pm 0.0523$ |
| ✓ | ✓ | ✓ | | $27.0598 \pm 2.1294$ | $0.7523 \pm 0.0297$ |
| ✓ | ✓ | ✓ | ✓ | $27.9796 \pm 2.6932$ | $0.8062 \pm 0.0247$ |

Transformer Module: Next, we evaluate the performance of the Transformer module alone. This module extracts global features by employing self-attention mechanisms to capture long-range dependencies and contextual information. the PSNR and SSIM of the reconstructed image are 26.1242 and 0.7091.

Feature Fusion: We evaluate the contribution of feature fusion by comparing the performance of our approach with and without the concatenated features from both the CNN and Transformer modules. The results are shown in row 3 of Table 4.

Channel Attention Module: We evaluate the effectiveness of the channel attention mechanism in enhancing the discriminative power of the fused features. This module selectively emphasizes informative channels while suppressing irrelevant or noisy channels within the fused feature representation.

Our ablation study confirms the effectiveness of each module and their combination for achieving high-precision tumor imaging in MPI images. The CNN module captures local features, the Transformer module captures global features, and the channel attention module enhances the discriminative power of the fused features.

### 3.3. Experimental Data Analysis

To improve the performance of our method on real-world datasets, we conducted transfer learning on a real dataset. Specifically, we fine-tuned the model for 20 epochs on the real dataset and then evaluated its performance by using two phantoms. Figure 7 represents the 3D model of phantoms. Figure 8 shows the visualization results obtained using different methods. After fine-tuning, our proposed method was capable of effectively

removing the blurring effect and restoring the concentration distribution of MNPs in the experiments data. These results demonstrate the effectiveness of our method in improving the quality of images obtained from realistic datasets. In addition to the complexity analysis, we have also conducted runtime calculations for different methods, as presented in Table 5.

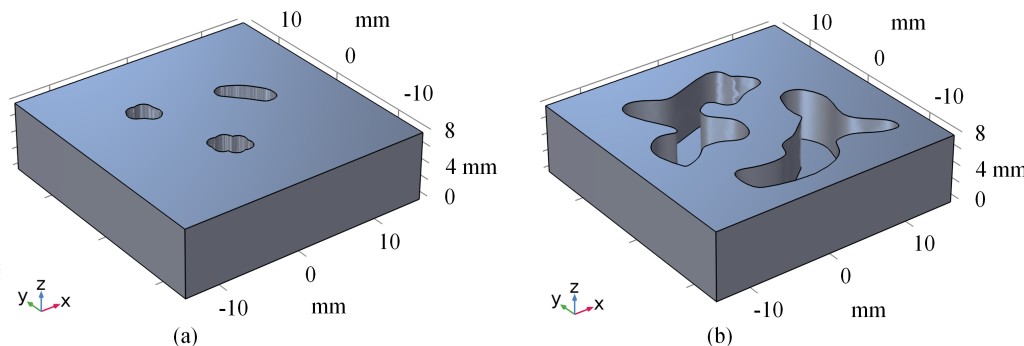

**Figure 7.** The 3D model of phantoms.

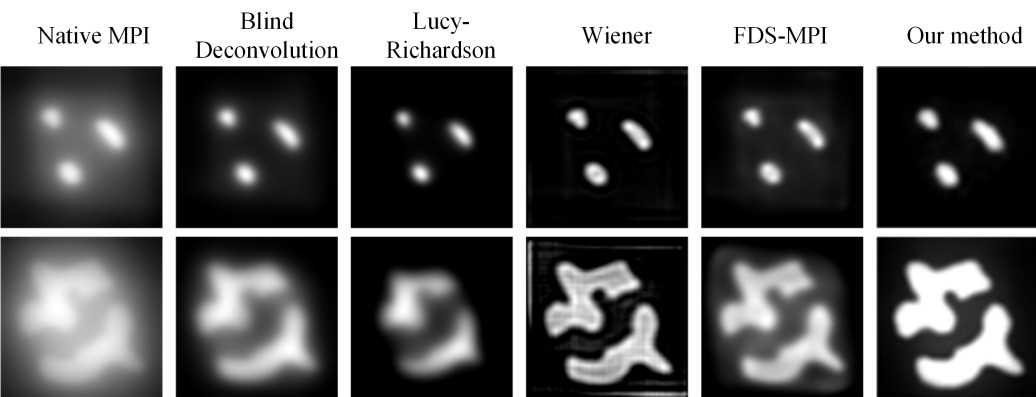

**Figure 8.** Visualization results of different methods on experiments data.

**Table 5.** The running time of different methods.

| Methods | Blind Deconvolution | Lucy-Richardson | Wiener | FDS-MPI | Our Method |
|---|---|---|---|---|---|
| Runtimes (s) | 0.1708 | 0.0472 | 0.0282 | 0.0203 | 0.0234 |

Moreover, to demonstrate the practicality of our proposed method, we conducted an in vivo experiment. Regarding the animal model, we used the C57BL/6 mice for the in vivo study. The study was conducted in compliance with IACUC-approved protocols. Anesthesia was induced using 4% isoflurane and maintained using 1–2% isoflurane during the imaging procedure. The MNPs was administered as a slow infusion via tail vein over a period of 1 min. The duration between injection and imaging was approximately 1 h. The imaging acquisition took approximately 2 min. The results are presented in Figure 9. Figure 9 shows (a) a white-light image of the mice, (b) a Native MPI image acquired by the commercial system, which exhibits significant blurring, (c) the result obtained by blind deconvolution, (d) the result obtained by Lucy-Richardson method, (e) the result obtained by Wiener method, (f) the result obtained by FDS-MPI, and (g) the result obtained by our proposed method. By applying the proposed method, we reduced the blurring around the region of interest, potentially leading to improved accuracy in identifying organs such as the liver and spleen. These findings further validate the effectiveness of our method in improving image quality for in vivo experimental applications. Our proposed method shows promise for enabling high-quality imaging of biological tissues and organs, which has important implications for biomedical research and clinical applications.

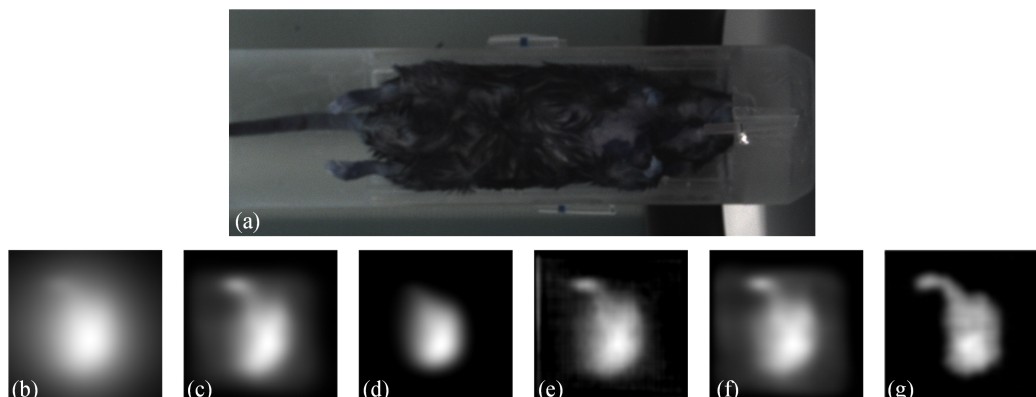

**Figure 9.** Visualization results on in vivo data. (**a**) white-light image of the mice. (**b**) Native MPI image. (**c**–**e**) the results of traditional methods. (**f**) the result of FDS-MPI. (**g**) the result of our method.

## 4. Discussion

When it comes to MPI image reconstruction, the x-space method commonly used in the field can result in blurry artifacts that make it difficult to accurately locate tumor lesions. This can significantly impact a doctor's ability to make informed decisions about a patient's condition. To overcome this challenge, we propose a deep learning-based method to precisely locate the tumor lesion. Our approach directly utilizes a network model to obtain clear MPI images. By leveraging both a CNN network and a Transformer network, we extract both global and local features from the MPI images, which is necessary for multi-target tumor imaging. Additionally, we fuse the features extracted by the CNN and Transformer networks and further extract useful features using channel attention mechanisms. Using our proposed method, we may help improve the quality of MPI images and achieve accurate imaging of tumor lesions. This has important implications for cancer diagnosis and treatment. By accurately locating tumor lesions, doctors can better assess the stage of cancer progression and determine the most appropriate treatment plan. Overall, our deep learning-based method represents an important step forward in improving the accuracy and effectiveness of MPI image reconstruction.

Table 4 demonstrates that each module plays a crucial role in restoring MPI image quality. The CNN module captures local features relevant to tumor localization, while the Transformer module captures global features, providing contextual information important for accurate localization. The channel attention mechanism further enhances the discriminative power of the fused features by selectively emphasizing informative channels. The feature fusion approach combining all modules achieves the best performance in terms of tumor localization accuracy, indicating the complementarity and synergy between the CNN and Transformer modules.

In order to simulate the clinical needs in real situations, we validate the imaging problem of low concentration targets under a unified field of view with multiple targets. As shown in Figure 6, we simulated the two tumors at different concentration ratios and added noise. When the concentration ratio is 1:0.2, the low concentration target is covered by noise, and the traditional deconvolution method is easy to identify the low concentration target as noise, so the reconstruction effect of the traditional method is poor. By taking into account the global characteristics of Transformer, our approach is able to restore low concentration targets, which is particularly important in clinical situations where there is interference from other organs such as the liver in actual imaging. Our method shows better robustness and accuracy in imaging low-concentration targets, which is related to its ability to capture global features and use channel attention mechanisms. In clinical applications, this ability can help doctors more accurately detect low-concentration lesions and improve diagnostic accuracy. It should be noted that in low concentration target imaging problems, noise and interference from other organs are common factors to consider. This may include factors such as body thermal noise and the interference caused by the liver's strong uptake

of MNPs affecting other imaging sites. Therefore, when designing the imaging algorithm, it is necessary to take into account the influence of these factors and adopt corresponding methods to reduce their influence. Our method provides a new idea and solution for low concentration target imaging, which can be used as an important reference for future clinical imaging technology.

In the experiment of transfer learning on real MPI images, we employed a pre-trained model for comparison. We selected a network trained on simulated data as the base model and fine-tuned it using the pre-trained model on the real MPI image dataset. We used the same hyperparameters and training strategy to ensure a fair comparison. The results showed that our network, after transfer learning, was able to handle real blurred data effectively. Additionally, in the future, we can explore extending this approach to other domains such as vascular imaging.

Our approach has several limitations. Firstly, our simulation model is based on the Langevin model which does not consider the relaxation of particles. This may result in reduced performance in handling particles with significant relaxation. In future work, we will consider the relaxation effect of particles. Secondly, our method only utilizes Native images in the X-space method and did not consider original signal information before X-space reconstruction. In future research, we can use multidimensional information (including images and signals) to optimize the reconstruction quality. Thirdly, while MPI is a functional imaging technique, it does not include structural information, which may lack information for accurate localization. In the future, we will explore multimodal imaging, such as merging MPI images with MRI images, to address this limitation. In addition, we will strive to further explore precise imaging of tumors within the body and address challenges encountered in in vivo tumor imaging, such as deep tissue imaging and dynamic change monitoring. Finally, we focused on evaluating the model's performance in accurately detecting and estimating MNPs concentrations without introducing additional complexity through concentration changes. In the future, we could explore the value of different MNP concentrations to model more diverse in vivo scenarios. This provides insight into how the model performs under different conditions and helps to further validate and improve the model.

## 5. Conclusions

In this study, we proposed a novel deep learning network model that combines a dual-branch convolution module, an efficient transformer module, and a feature fusion with channel attention module. Furthermore, our proposed method successfully mitigates blurring artifacts, leading to improved image quality and enhanced spatial localization capabilities. Our results confirm the superiority of our proposed method over traditional approaches. The clearer imaging achieved through our approach has the potential to significantly impact the field of MPI by improving spatial precision and accuracy.

**Author Contributions:** Conceptualization, Y.S.; Formal analysis, Y.S.; Funding acquisition, J.L.; Investigation, Y.S.; Methodology, Y.S.; Project administration, J.L. and Y.W.; Resources, J.L. and Y.W.; Software, Y.S.; Writing—original draft, Y.S.; Writing—review and editing, J.L. and Y.W. All authors have read and agreed to the published version the manuscript.

**Funding:** This research was funded by the National Natural Science Foundation of China grant number KKA309004533 and 81571836. Beijing Jiaotong University Science and Technology Foundation of grant number 2006XM006, JS2002J0160 and JS2002J0080.

**Institutional Review Board Statement:** The animal study protocol was approved by the Institutional Animal Care and Use Committee of Institute of Automation, Chinese Academy of Sciences (protocol code IA21-2302-420106 and date of approval 6 March 2023).

**Informed Consent Statement:** Not applicable.

**Data Availability Statement:** The data presented are openly available in https://github.com/kutuy123/MPI-Phantom-dataset.

**Conflicts of Interest:** The authors declare no conflict of interest.

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
