# Peer review of "Fusion Networks of CNN and Transformer with Channel Attention for Accurate Tumor Imaging in Magnetic Particle Imaging"

_biology, doi:10.3390/biology13010002_

Round 1

Reviewer 1 Report

Comments and Suggestions for Authors

Overall a good article with 3 major questions:

1. Is there a connection with the previous work of the authors where they worked on improving the resolution component?

2. The authors mention multimodality via MRI, for future work, but within the presented framework, what expected complexity is expected when dealing with issue beyond localization accuracy?

3. It is totally missing the aspect of reproducibility of this work, no links appear to methods, data, code.

Reviewer 2 Report

Comments and Suggestions for Authors

Dear Dr. Thanasomboon and Dear authors,

Thank you for the opportunity to review this manuscript by Shang et al., entitled "Fusion Networks of CNN and Transformer with Channel Attention for Accurate Tumor imaging in Magnetic Particle Imaging". This work proposed a novel deep learning-based approach to reconstruct MPI data. The proposed DL model was trained and validated using simulated data, and was subsequently applied on 3D-printed phantom and in vivo images, where it outperfromed existing reconstruction algorithms.
We find this subject novel enough, and could attract audiences interested in technical development and in vivo applications of MPI. That being said, rigor & presentation of the approach, results and discussion sections all left something to be desired. In particular, details were often missing and should be expanded.

Our specific comments are as follows,

1. Page 1 line 2: "traditional imaging methods often struggle to precisely locate tumors, resulting in challenges when quantifying them." This is not true. Conventional cross-sectional imaging such as CT/MRI and molecular imaging modalities like PET work reasonably well with tumor identification and localization. Perhaps the authors meant tumor localization is challenging using existing MPI acquisition/recon approaches?

2. Page 1 line 9, Page 11 line 281, Page 14 line 344, etc: "...improving the precision of tumor localization", "...facilitating precise tumor localization", "... achieve accurate imaging of tumor lesions." These statements seemed overreaching as no in vivo tumor data was presented, and it cannot be directly concluded that the accuracy of tumor detection would be improved. Saying something like "The data suggest the proposed approach may help improve tumor detection & localization in the future..." would have been better.
Additionally, it should be made clear that the intro paragraph "... we send the tumor mice into the MPI platform" and Figure 2 schematic represented the idea how MPI works, rather than what has actually been done in this study, as it didn't appear any tumor-bearing mice was scanned.
Similarly, the term "tumor" was kept being brought up in the results, such as in Figure 5, making it confusing as of what the nature of these images was.

3. Page 1 line 26: The conclusion was somewhat vague. "... significant implications for the advancement and clinical application of MPI technology". The same can be said about any other advancements in MPI research. The conclusions should be more specific to the novelty and the findings of this particular study.

4. Page 4 line 121: Much detail was missing from the simulation methods. It wasn't clear how the 8000 numerical phantoms were created. What kind of imaging features were designed in the numerical phantom, and what parameters were used? e.g. How many features/lesions were present in each image/FOV, and what were their shape, size, geometry, contrast-to-background ratio, etc?
A few parameters of the signal equation were given in Table 1, but the signal equation and/or the point spread function for this particular imaging system to create the simulated data was nowhere to be found.
If the simulation approach was previously described, proper citation should be provided. Otherwise, it should be described in detail.

5. Page 5 line 152: "The MNPs were inserted into the phantoms at a concentration of 1 mg/ml." Was there a particular reason why the same concentration of MNP solution was dispensed into the multi-compartment phantom? Wouldn't having different MNP concentrations in each compartment better reflect in vivo MNP distribution, and thus improving model performance?

6. Page 5 line 146-155: Similar to the digital phantoms, design of the 3D printed phantoms should be better described. e.g. feature size, shape, pattern, etc.

7. Page 5 line 124-145: The paragraph appears subjective with multiple uses of terms such as "thoughtfully", "powerful", "solid principles of ...", "skillfully","meticulously". To stay objective, it is generally recommended to avoid this type of language in scientific writing. Ditto for Line 163 "well-trained model". Just "trained model" would suffice.

8. Page 5 line 140-144: "intricate nuances of noise patterns" is again quite vague and subjective. It wasn't clear in which sense the noise pattern was "intricate." What was the statistical distribution of noise -- zero-mean Gaussian, Rician, Poisson or pink? Was the noise spatially varying or spatially invariant?
Rather than a qualitative description of "diverse signal-to-noise ratios (SNR)," the actual, quantitative SNR range used in simulation should be provided. Were these SNR numbers consistent with the scanned phantom and in vivo data?

9. Page 5 line 159: "... augmentation operations, including inversion and rotation, to enrich its diversity." Inversion and rotation are rigid transformations, after which a lot of the image features (e.g. feature size, shape, signal intensity) remain the same. How would this be expected to diversify the data?
Similarly, (line 142.) "... we introduced an element of randomness by cropping these captured noise images." The cropped image is a subsample of the original, uncropped images. Their noise distribution would be fully correlated, and is therefore not random.

10. Page 4 line 98: "... across both phantom and physical environments" What does physical environment mean?

11. Page 6 line 165-174: The model description here were duplicates from the 5th paragraph of the intro (around line 102). The paragraph seems better suited in the methods, and therefore that in the intro should be removed or abbreviated.

12. Page 10 line 159: "To address this issue, we propose a novel deep learning algorithm that enables simultaneous reconstruction of two tumor targets with different concentrations." Is this a part of the overall proposed framework? The wording seems ambiguous here and no description could be found how the framework deals with features with different signal intensities.

13. What was the effective point spread function using the proposed DL recon? Can it be characterized, e.g. by imaging a point source, and how does it compare to those from the existing recon methods?

14. Page 12 line 293: "table 4". "T" should be capitalized.

15. Page 12 line 312: "We conducted transfer learning ..." Transfer learning was mentioned both in the results and discussions. However, how exactly "transfer learning" is done was missing from the methods and should be added.

16. Page 13 line 320-332: In vivo rodent data was presented in this section. However, approach for the in vivo study was missing from the methods section. e.g. What was the breed of the animal? Was the study in compliance with IACUC-approved protocols? How was the anesthesia done? Was the MNP given as a quick IV push, or a slow infusion, over how long? What was the duration between injection and imaging, and how long did the image acquisition take? What were the FOV size, resolution, and other relevant scan parameters?
Additionally, it would be informative to provide the typical CNR & SNR in vivo to benefit future simulation and phantom experimental designs.

17. Page 13 line 327: "... removes the blurring around the liver while simultaneously preserving the realistic shape of the liver and spleen." It is rather challenging to discern the liver and spleen from the MPI image alone. Were there any anatomical reference scan acquired with CT or  T1/T2-weighted MR? Else perhaps considering using more speculative rather than assertive language?

18. Page 14 line 349: "... results demonstrate that each module plays a crucial role in achieving high-precision tumor localization". "... The feature fusion approach combining all modules achieves the best performance in terms of tumor localization accuracy." These statements again seem overreaching. To determine whether the combination of the two modules performs the best, it should be compared to the performance of each module alone, which was not presented in the results.

19. Page 14 line 365-372: "Our method shows better ... reduce their influence". The whole paragraph reads weird. Does "noise and interference from other organs" mean body thermal noise, or signal due to nonspecific uptake of MNP by intestinal organs? The former can benefit from the improved algorithm to "reduce their influence", whereas the latter should be accurately reconstructed as it reflects physiology. This needs to be better clarified.

20. Page 14 line 369: "accuracy and accuracy". The word accuracy is duplicated.

21. Page 14 line 386: "... our method only utilizes Native images in the X-space method and did not consider signal information." This sentence is overall unclear. What does "signal information" mean?

22. Table 2 and 3: Supposely a few thousands of numerical and hundreds of physical phantom datasets were acquired. In such case, standard deviation should be provided for metrics like SNR and SSIM in addition to the mean value. Additionally, the labels for 1:0.2, 1:0.4... should be added to Table 3. Similarly, the rows and checkmarks in Table 4 are difficult to interpret without labels.

Best regards,

Comments on the Quality of English Language

While NLP-assisted writing is good, it should be proofread by human to ensure adherence to scientific writing styles.

Reviewer 3 Report

Comments and Suggestions for Authors

The following points need to be addressed in the revision:

  1. The main innovation and contributions of this research should be clarified in a bulleted form in the introduction at its end.
  2. The structure of the article is missing that need to be mentioned at the end of the introduction.
  3. Comparison with SOTA techniques should be mentioned in the abstract.
  4. The section related to the dataset is not clear. Is the dataset publicly available or not? If you are simulating the data, then would it be available to the community for checking the reproducibility of your work?
  5. Can you compare the results with similar studies or some previous findings of similar studies?
  6. Figure 3 needs to be improved in many places. The involvement of your dataset, along with the usage of CNN and transformers is wrong and needs to be improved again. The involvement of evaluation metrics is not clear.
  7. In Figure 3, I noticed some terms that have never been discussed in the entire article, like SPIONs, etc.
  8. The flow of the data is wrong, and it needs to be corrected in Figure 3.
  9. The architecture of Figure 4 is not acceptable and needs to be thoroughly improved in multiple places. The details related to the labeling of the architecture should be reinvented and discussed in the article.
  10. Please make the source codes and dataset available to increase the visibility of this manuscript.
  11. The evaluation metrics section is missing citations.
  12. Many typos and grammatical mistakes in the article need to be corrected.
  13. Your study is lacking of complexity analysis of the algorithms. Please add the validation results in your work with probability-based time and computational analysis of your work.
  14. So many terms without any reference need to be appropriately cited.
  15. A more direct outcome of the study and its possible applications may be included.
  16. Figure 5 is lacking of spatial resolution. Include the details at the appropriate position in the article. Also, add these details in the caption of Figure 5.
  17. Revise your conclusions section based on the modifications suggested throughout the article.
  18. I think the channel attention module was never discussed in the article irrespective of the fact that it seems to be an important part of Figure 3.

Comments on the Quality of English Language

The article needs to be improved.

Round 2

Reviewer 1 Report

Comments and Suggestions for Authors

ok, I recommend you specify in the paper that the reproducibility of the methods will be made available upon publication through a link in the paper.

Comments on the Quality of English Language

minor

Author Response

Thank you for your suggestion. We appreciate your recommendation, and we will make the reproducibility of our methods available to the scientific community after the publication of the paper.

To address the language issues, we sought the assistance of researchers within our field to review and provide feedback on the manuscript. We carefully implemented their suggestions and made appropriate revisions to enhance the clarity and coherence of the paper.  

Reviewer 2 Report

Comments and Suggestions for Authors

Dear Dr. Thanasomboon and dear authors,

Thanks for preparing this revision which has led to improvements from the original manuscript. Here are a few lingering issues that should be addressed prior to publication.

1. Page 1 line 2: "traditional imaging methods often struggle to precisely locate tumors, resulting in challenges when quantifying them." As was mentioned in our prior comments, this sentence should be revised since the term "traditional" or "conventional" imaging refers to CT/MRI (as opposed to new molecular imaging methods). The statement would otherwise be incorrect.

2.  Page 5 line 140: In the signal equation, is "r" a vector? If so, it should either be in bold typeface or have an accent on top.

3. Page 5 line 158: It appears that a lot of key quantitative measures are still missing from the approach. For instance, "diverse signal-to-noise ratios (SNR)" is not as informative as saying, "SNR ranging from 10 to 100." Same goes for "different ratios". Saying something like "10 different ratios were chosen, ranging from x to y." would improve clarity.

Additionally, the statistical distribution of the noise was still not described nor cited.

4.  Page 5 line 164: "The shape of the features is randomly generated" is still rather vague and may adversely impact the reproducibility of this research. What was the algorithm used to generate these "random" features? What were their size, shape, intensity specifications? For instance, the digital phantoms shown in Figures 5 and 6 appeared to have specific patterns, which would not have resulted from randomly assigning a binary value to each voxel within the FOV. How exactly were they created?
Again, citations should be provided if this has been previously described. Otherwise, it should be described in more detail.

5. Page 13 line 164: "Regarding the animal model, we used Sprague-Dawley rats ...." This section belongs to the methods rather than the results. A discrepancy now exists between the text which claims Sprague-Dawley rats was used vs Figure 9 captions saying "mice". This needs to be reconciled.
On a related note, the said rat model is normally substantially larger than what a 4x4cm scanner and FOV can fit. Perhaps it should be revisited exactly what animal model was used, or whether the FOV measurement was accurate?

6. Supporting information should be properly cited in the relevant sections of the main text.

Best regards,

Comments on the Quality of English Language

N/A

Reviewer 3 Report

Comments and Suggestions for Authors

All my points have been thoroughly addressed.

Author Response

Thank you for your feedback. We are pleased to hear that all your points have been thoroughly addressed.